# Product Validation and Stability Testing of Pharmacy Compounded Cholic Acid Capsules for Dutch Patients with Rare Bile Acid Synthesis Defects

**DOI:** 10.3390/pharmaceutics15030773

**Published:** 2023-02-26

**Authors:** Yasmin Polak, Bart A. W. Jacobs, Natalja Bouwhuis, Carla E. M. Hollak, Maurice A. G. M. Kroon, Elles Marleen Kemper

**Affiliations:** 1Department of Pharmacy and Clinical Pharmacology, University of Amsterdam, 1105 AZ Amsterdam, The Netherlands; 2Medicine for Society, Platform at Amsterdam UMC, University of Amsterdam, 1105 AZ Amsterdam, The Netherlands; 3Department of Endocrinology and Metabolism, Amsterdam UMC, University of Amsterdam, 1105 AZ Amsterdam, The Netherlands

**Keywords:** cholic acid, pharmacy compounding, bile acids, orphan drugs, product validation, stability testing, good manufacturing practice (GMP), active pharmaceutical ingredient (API), bile acid synthesis defects (BASD)

## Abstract

Bile acid synthesis defects (BASDs) comprise a group of rare diseases that can be severely disabling. Bile acid supplementation with 5 to 15 mg/kg cholic acid (CA) has been hypothesized to decrease endogenous bile acid production, stimulate bile secretion, and improve bile flow and micellar solubilization, thereby improving the biochemical profile and potentially slowing down disease progression. Currently, CA treatment is unavailable in the Netherlands, and CA capsules were compounded by the Amsterdam UMC Pharmacy from CA raw material. This study aims to determine the pharmaceutical quality and stability of the pharmacy compounded CA capsules. Pharmaceutical quality tests were performed on 25 mg and 250 mg CA capsules according to general monographs of the European Pharmacopoeia 10th ed. For the stability study, the capsules were stored under long-term conditions (25 °C ± 2 °C/60% ± 5% RH) and accelerated conditions (40 °C ± 2 °C/75% ± 5% RH). Samples were analyzed at 0, 3, 6, 9 and 12 months. The findings demonstrate that the pharmacy compounded CA capsules within a range of 25–250 mg that complied with the European regulations in regard to product quality and safety. The pharmacy compounded CA capsules are suitable for use in patients with BASD, as clinically indicated. With its simple formulation, pharmacies are provided a guidance on product validation and stability testing when commercial CA capsules are unavailable.

## 1. Introduction

Bile acid synthesis defects (BASDs) are a group of rare diseases either due to single enzyme deficiencies (SEDs) in bile acid synthesis or Zellweger spectrum disorder (ZSD), in which there is disruption of peroxisomal biogenesis [1]. All can be severely disabling [1,2,3,4]. It is unknown which biochemical abnormalities exactly contribute to the pathophysiology of the different types of BASDs (i.e., liver disease—usually presented during childhood—and neurological disease—presenting during later childhood or later adult life) [3]. Reduced canalicular bile acid excretion and abnormally elevated levels of toxic C27-bile acid intermediates are thought to contribute to liver disease [1,4,5,6,7]. Abnormally elevated bile acid intermediates are thought to be toxic for the brain [8,9], potentially causing neurological disease. Despite the paucity of clinical evidence, bile acid supplementation with 5–15 mg/kg cholic acid (CA) for the treatment of different types of BASDs has been available for many years [10,11,12,13,14,15,16,17,18]. It has been hypothesized that cholic acid supplementation decreases endogenous bile acid production, stimulates bile secretion, and improves bile flow and micellar solubilization, thereby improving the biochemical profile and potentially slowing down disease progression [2,7,19,20].

The orphan medicinal product Orphacol^®^, which holds CA as the active ingredient, is currently the only licensed medicinal product for the treatment of specific subgroups of BASD patients in the European Union (EU), namely 3β-Hydroxydelta-5 -C27-steroid oxidoreductase (3β-HSD) deficiency and delta-4 -3-Oxosteroid-5β-reductase (5β-reductase) deficiency [21,22]. In the United States (U.S.), another CA medicinal product with orphan drug designation is registered under the name Cholbam^®^. Cholbam^®^ is registered for a broader spectrum of BASDs, namely BASD due to any type of SED [23]. Additionally, Cholbam^®^ can also be given as an adjunctive treatment of peroxisomal disorders (PDs), including ZSD, in patients who exhibit manifestations of liver disease, steatorrhea, or complications from decreased fat-soluble vitamin absorption [23]. Despite the registration of Cholbam^®^ for this broad spectrum of indications in the U.S., the European variant of Cholbam^®^—Kolbam^®^—was only granted EU market authorization for three specific SEDs: sterol 27-hydroxylase deficiency (cerebrotendinous xanthomatosis, CTX), alpha-methylacyl-CoA racemase (AMACR) deficiency, and cholesterol 7-alpha-hydroxylase (CYP7A1) defect [24]. However, the marketing authorization for Kolbam^®^ was withdrawn in July 2020 at the request of the marketing authorization holder, Retrophin Europe Ltd. [25].

Currently, CA treatment is inaccessible in the Netherlands. Orphacol^®^, the only EU-authorized CA treatment, is expensive and has not been made available in the Netherlands. Although there is still uncertainty about the clinical effect and appropriate use of CA, access for Dutch BASD patients in a controlled setting provides the opportunity to evaluate its effects. Moreover, Orphacol^®^ only exists in 50 mg and 250 mg capsules [21]. Practice has shown that for children receiving a daily dose of 5 to 15 mg/kg [21], dose adjustment is desired in order to obtain the correct dosage or improve ease of use. A safe and sustainable alternative to the expensive EU-registered medicinal product can be achieved through pharmacy compounding from CA raw material. We developed a pharmacy formulation for CA capsules that could be easily made by pharmacists with compounding facilities [26]. The aim of the study is to determine the pharmaceutical quality and stability of our pharmacy compounded CA capsules. 

## 2. Materials and Methods

### 2.1. Chemicals and Materials

CA active pharmaceutical ingredient (API) was purchased from a trusted GMP manufacturer. The excipients colloidal silica and lactose monohydrate comply with Ph.Eur. quality standards and were purchased from Spruyt Hillen (Capelle aan den Ijssel, The Netherlands). The capsules (sizes 0 and 3) used are 100% hard gelatin, compliant with the European Pharmacopoeia (Ph.Eur.) quality standards and were purchased from Spruyt Hillen. The reference standards for cholic acid CRS (CA-CRS) (C2158000), chenodeoxycholic acid CRS (CDCA-CRS) (C1050000), litocholic acid CRS (LCA-CRS) (L0720800), and ursodeoxycholic acid (UDCA-CRS) (U0800000) were Ph.Eur. grade reference standards purchased from the European Directorate for the Quality of Medicines and HealthCare (EDQM, Strasbourg, France). For deoxycholic acid (DCA), methyl-cholic acid (MCA), and other chemicals, the reference standards were of analytical grade and were purchased from Sigma Aldrich. 

### 2.2. Preparation of Cholic Acid Capsules

The Amsterdam UMC pharmacy holds a GMP license for the manufacturing of investigational medicinal products (packaging, labeling, and manufacturing of capsules). Cholic acid capsules were prepared by the Amsterdam UMC Pharmacy (Pharmacy AMC) by manual physical mixing (mortar and pastle). For 250 mg capsules, size 0 hard gelatin capsules were filled with a powder mixture of 250 mg CA API and colloidal silica 1% (*w*/*w*) (lubricant). For the 25 mg capsules, size 3 hard gelatin capsules were filled with a powder mixture of CA API, colloidal silica 1% (*w*/*w*), and lactose monohydrate (filler) (total volume of 27 mL/100 capsules). Three batches of 1200 and 1300 capsules each were produced at 250 mg and 25 mg strength, respectively, in partial portions of 100 capsules at a time. In order to limit inter- and intra-batch variations, an in-process control on weight was performed on each portion of capsules, with an allowed variation of no more than 4% (n = 10 capsules per portion). The capsules were packed as follows: 25 capsules per each white pharmaceutical HDPE Duma^®^ Twist-Off 100 mL container with a Duma^®^ Twist-Off cap.

### 2.3. Pharmaceutical Quality Tests

Appearance, identification through infrared spectrophotometry (IR; using IRAffinity-1S from Shimadzu), uniformity of mass of single dosage units (N = 20), uniformity of content, content assay, loss on drying (LOD), dissolution, disintegration, and microbiological tests were performed according to the general monographs of the Ph. Eur. 10th ed. [27,28,29,30,31,32,33]. See Table 1 for the test specifications. 

### 2.4. High-Performance Liquid Chromatography Refractive Index

Concentrations of CA and related substances (CDCA, DCA, MCA, or unidentified) were determined using a high-performance liquid chromatography system with refractive index (HPLC-RI) detection using RID-10A from Shimadzu. A reversed phase X-bridge BEH phenyl column measuring 250 × 4.6 × 3.5 μm (Waters Corporation) was used. Column temperature was kept at 30 °C ± 2 °C. All samples were dissolved and diluted in methanol and subsequently analyzed. The reference and sample solutions were tested to be stable for 7 days at room temperature (15–25 °C) and in the refrigerator (2–8 °C). The mobile phase consisted of 30 mmol/L potassium dihydrogen phosphate (KH2PO4) and 600 μL phosphoric acid in a 1 L mixture of water/acetonitril (ACN) (60:40). The differential refractometer was maintained at 40 °C. The injection volume was 30 μL. The flow rate was 1.0 mL/min. The chromatograms were obtained with a resolution of ≥1.5. The retention times were ±8.3 min for CA, ±17.8 min for CDCA, and ±21 min for MCA. Chromatograms were processed using Labsolutions software from Shimadzu. The HPLC-RI method was validated according to the Ph.Eur. individual monograph for CDCA [34], which described an analytical method similar to what was required and the ICH Q2(R1) guideline regarding accuracy, precision, specificity, detection and quantitation limits, and linearity [35].

### 2.5. Stability Study

Three batches were produced of 25 mg and 250 mg capsules. The capsules were subjected to a stability program of 12 months under room temperature conditions (25 °C ± 2 °C/60% ± 5% RH) and 6 months under accelerated conditions (40 °C ± 2 °C/75% ± 5% RH) according to the ICH Q1(R2) guideline [36]. The capsules were analyzed for appearance, identity, content, content uniformity, loss on drying, related substances, microbiological quality, and disintegration and dissolution profiles at 0, 3, 6, 9, and 12 months.

### 2.6. In Vitro Dissolution Testing

Dissolution tests were performed using a type II paddle with sinker dissolution apparatus (PharmaTest, Germany) as described in the FDA dissolution methods database [37] at a rotation speed of 100 rpm. At 37 °C ± 0.5 °C, one 25 mg or 250 mg capsule was placed in 500 mL or 900 mL phosphate buffer (pH 6.8) respectively [38]. Samples of 5 mL were taken at T = 5, 10, 15, 20, and 30 min through a 0.45 μm PVDF filter and measured using the validated HPLC-RI system.

## 3. Results

### 3.1. Productvalidation

The mean capsule weight was 209.9 mg and 251.0 mg for the 25 mg and 250 mg capsules, respectively (Table 2). For both strengths, all capsules in the three batches had a mass variation below 10% (n = 20), and the batches had a mean acceptance value (AV) of 10.1 and 7.6 (AV ≤ 15), respectively, for the 25 mg and 250 mg capsules (Table 3). The 25 mg capsules disintegrated within 6 min, and the 250 mg capsules within 9 min. For both strengths, full dissolution was seen at 10 min, with an average dissolution of 102.5% for the 25 mg capsule and 100.4% for the 250 mg capsule (Figure 1). The total amount of related substances was not more than 0.10% for all three batches of 25 mg, and no more than 0.14% for all three batches of 250 mg (the maximum allowed amount is 1.00%). The mean result for loss on drying was 0.8% and 0.2% for the 25 mg and 250 mg capsules, respectively (the maximum allowed amount is 5.0%). For the 25 mg capsules the microbiological results were 10 CFU/g for TAMC and <5 CFU/g for TYMC. For the 250 mg capsules the microbiological results were <5 CFU/g for TAMC and <5 CFU/g for TYMC. In all batches, *E. coli* was absent (Table 3).

### 3.2. Stability Study

#### 3.2.1. Identification and Content Assay

At all time points, the 250 mg CA capsules meet the stability criteria for identification and content (Table 4). For the 25 mg CA capsules, all time points meet the specification, with the exception of two of the three 25 mg CA batches at T3, where the measured CA content was below the specification limit The mean content values at this time point were 89.9% ± 3.89 and 88.6% ± 3.81 for the capsules stored under long-term conditions (25 °C ± 2 °C/60% ± 5% RH) and accelerated conditions (40 °C ± 2 °C/75% ± 5% RH), respectively. A standard out of specification (OOS) investigation showed that the recovery for this test was too low, plausibly resulting in low content measurements. Subsequently, the 25 mg CA content measurements for T0 and T3 were declared invalid (shown in Table 4 as ‘NA’ (not applicable)). Following the OOS investigation and conclusions, the method was further optimized, and sufficient recovery was achieved. 

#### 3.2.2. Related Substances

As seen in the product validation, the only related substance discovered was MCA, the methyl ester of cholic acid (Table 4). No CDCA, DCA, or unidentified related substances were found in the capsule samples. The measured MCA percentage exceeded the specification limit (≤0.20%) in the 250 mg CA capsule at 3 months under both storage conditions. After three months, 0.27% of MCA was discovered under long-term storage conditions, and 0.23% of MCA was discovered under accelerated storage conditions (Table 4). At all other time points, MCA percentage fell under the specification limit of 0.20%, and storage duration/condition did not seem to have a structurally significant effect on the increase of MCA presence, or the occurrence of any other related substance. 

#### 3.2.3. Dissolution and Disintegration of CA Capsules

The storage period, or the condition under which the capsules were stored, had no significant impact on the dissolution and disintegration strength profiles of CA capsules. A trend was visible in the dissolution profile of the capsules. Capsules which were stored for a longer time, took more time to dissolve and to release its contents. The storage condition had an effect on the dissolution profile as the capsules stored under accelerated storage conditions (40 °C ± 2 °C/75% ± 5% RH) showed a greater shift in the dissolution profile compared to capsules stored under long-term storage conditions (25 °C ± 2 °C/60% ± 5% RH). This shift in dissolution profile seemed more prominent for the 25 mg CA capsules compared to the 250 mg CA capsules (Appendix A). However, the dissolution times did not change significantly over time, and the specification limit was amply met each time, regardless of capsule strength or storage condition (Appendix A).

After 3 months of storage, the disintegration times of both capsule strengths decreased, with a mean disintegration time of 5.67 and 8.33 min for the 25 mg and 250 mg CA capsules, respectively, at T0 and a mean disintegration time of 2.00 and 3.00 min, respectively, at T3. Longer storage did not have an effect on significantly decreasing the disintegration time any further. In addition, the storage condition did not have a further significant effect on the disintegration time (Appendix A). Moreover, the specification limit for the disintegration time (≤30 min) was easily met every time.

### 3.3. Microbiology

For all batches at all time points, the microbiological quality met the specifications well below the limits given for TAMC and TYMC. The highest value found for TAMC was 10 CFU/g for the 25 mg CA capsules and below 5 CFU/g for the 250 mg CA capsules. For TYMC, values fell below 5 CFU/g for all batches. *E. coli* was absent in all batches. Storage condition or duration had no significant effect on microbiological growth.

## 4. Discussion

Patients with rare diseases are often confronted with the fact that effective medicines are unavailable or not yet developed. This jeopardizes the health of vulnerable patients with rare diseases. An additional challenge is that medicines for rare diseases come to market with limited evidence for effectiveness in a real-world setting. Hence, there may be a need for additional investigation to establish their appropriate use. For CA as a therapeutic treatment for patients with BASD, there is a combination of limited real-world evidence and unavailability [11,12,13,16,17,18,39,40]. Pharmacy compounding can provide a solution by producing a medicine for a patient with a specific need that cannot be met with an authorized medicine that is available on the market. However, pharmacy compounding also provides a solution in case an authorized medicine is not available at all. 

Just like EU commercial medicinal products, pharmacy preparations have to comply with EU regulations in terms of quality and product safety, whose specifications are stated in the European Pharmacopoeia (Ph. Eur.). Our findings demonstrate that the pharmacy compounded CA capsules within a range of 25–250 mg complies with the EU regulations in regard to quality and product safety [26]. During the product validation, the capsule content showed some larger variations within one batch in comparison with others (e.g., batch 3, with an AV of 14.7). This is due to the fact that the capsules are manually filled per 100 capsules at a time, essentially creating sub-batches, and therefore one sub-batch of 100 capsules can show variation from another sub-batch. Nonetheless, all content measurements fall within the specification of 90–110% of labelled content specification. During the stability study, CA contents were measured slightly below 90% at 3 months for 2 out of 3 batches of the 25 mg CA capsules. A standard OOS investigation determined that an incorrect method for the sample preparation was performed, which resulted in a low recovery due to decreased solubility of CA (as a result of adhesion to lactose monohydrate (filler)), and hence plausibly low content measurements. All T0 and T3 results for the 25 mg CA capsules were subsequently declared invalid. The method was further optimized by adding more diluent while keeping the final CA concentration unchanged. This adjustment to the sample preparation ensured CA would go into complete solution again, improving recovery. All further content measurements were within the specification limits of 90–110%. 

The product validation and stability studies show that MCA is an impurity that is always present in the capsules. This impurity originates from the CA raw material, for which the manufacturer also reports this impurity on the certificate of analysis. In the stability study, MCA contents above the specification limit of 0.20% were measured at 3 months in the 250 mg CA capsules stored under long-term conditions as well as accelerated conditions. MCA contents fell well below the specification limit again in all following measurements at 6, 9, and 12 months (for both storage conditions), with similar concentration levels as found in the CA raw material. It is curious how it is possible that the MCA concentration is lowered again at later time points. Since the MCA concentration levels found in the CA capsules are similar to the MCA concentration levels reported for the CA raw material, it is not assumed that MCA is a degradation product that can be formed during storage. The increased MCA levels at 3 months are most likely due to an incorrect analysis performance at this time point, and that the reported results are invalid. However, this would have to be analyzed further to be certain. Nonetheless, study in patients found that CA preparations with MCA concentrations of 0.4–0.5% caused no adverse events when given therapeutic doses of CA [41]. Therefore, it is possible to broaden the specification limit for MCA. This is not uncommon, as specification limits can be adjusted upon further investigation, taking the daily ingested amount and its potential toxicity into consideration. 

As for dissolution, no specific requirement has been set for regular hard gelatin capsules. However, it is evident that the capsule content has to be dissolved in order for the active ingredient to be absorbed and have a systemic effect. Our study showed that maximum dissolution is seen at 5–10 min, which allows for rapid absorption of CA.

In addition, the quality of the pharmacy compounded CA capsules has been shown to be stable (also under accelerated storage conditions) for a longer period of time, allowing them to be kept in stock and adequately stored in the patients’ home. Based on this product validation and stability study, both adult and pediatric patients with BASD can be studied at any desired dose ranging from 25 to 250 mg, an aspect that is not possible with commercially available capsules. 

The highest and lowest capsule strengths were analyzed for this study, representing the highest and lowest active substance/excipient ratios as well as the highest and lowest risk formulations. The results show a range of the pharmacy formulation’s extremes and limitations. It is assumed that all strengths between 25 and 250 mg fall between these extremes and limitations in terms of content, content uniformity, impurities, microbiological quality, and disintegration and dissolution profiles. However, additional analyses of intermediate strengths would have to be performed to validate whether they indeed fall within the range of extremes and limitations seen in this study.

Currently, the pharmacy compounded CA is only provided in a clinical study setting [42]. However, the pharmacy CA preparation could also provide a sustainable and affordable alternative for CA treatment for Dutch BASD patients in regular healthcare in the future, as long as a commercial product is unavailable. An advantage of the pharmacy CA capsules is that the dose can easily be adapted to the dosage requirements of the patient, improving the ease of use and limiting the risk of dosing errors. In addition, the pharmacy compounded CA is making quick dose adjustments based on dose-response (i.e., biochemical response and side effects) more feasible. 

## 5. Conclusions

In conclusion, we developed CA capsules suitable for use in clinical studies regarding patients with BASD. With this simple formulation, we provide pharmacies with guidance on product validation and stability testing when commercial CA capsules are unavailable. The stability data in this study show that the pharmacy compounded capsules are stable, also in the case of temperature excursions, and that a shelf life of 12 months can be given.

## Figures and Tables

**Figure 1 pharmaceutics-15-00773-f001:**
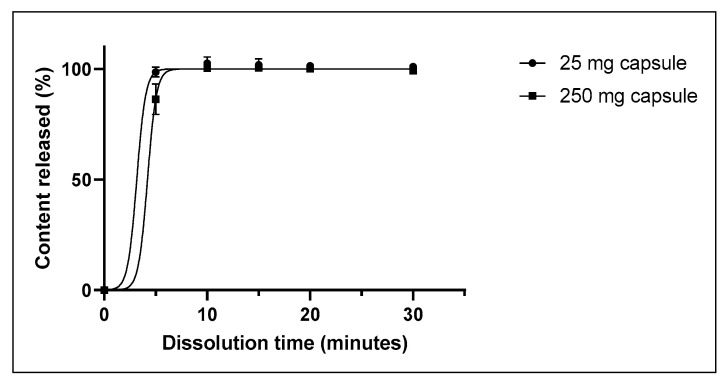
Dissolution profiles of 25 mg and 250 mg CA capsules (mean ± SD, n = 3).

**Table 1 pharmaceutics-15-00773-t001:** Pharmacotechnical test specifications for compounded cholic acid 25 mg and 250 mg capsules.

Test	Test Reference	Specification	Specification Reference
Appearance	In-house	White or almost white powder	In-house
Identification, IR, and HPLC	Ph.Eur. 2.2.24 (IR) In-house (HPLC)	Positive with reference standard	Cholic acid Ph.Eur. CRS
Content (% of labelled content)	In-house	90.0–110.0	In-house
Content uniformity	Ph.Eur. 2.9.40		Ph.Eur. 2.9.40
Mass variation ^†^ (%)		≤10.0	
Content uniformity * (AV)		≤15.0	
Loss on drying (%)	Ph.Eur. 2.3.32	≤5.0	In-house
Related substances (%)	In-house		In-house
CDCA		≤0.15	
DCA		≤0.15	
MCA		≤0.20	
Any unidentified substance		≤0.10	
Total unidentified substances		≤0.40	
Total Related Substances		≤1.00	
Dissolution (minutes) ^±^	Ph.Eur. 2.9.3 and FDA [37]	<30	In-house
Disintegration time (minutes)	Ph.Eur. 2.9.1	<30	In-house
Microbiology	Ph.Eur. 2.6.12 Ph.Eur. 2.6.13		Ph.Eur. 5.1.4.
TAMC (CFU/g)		<1000	
TYMC (CFU/g)		<100	
*E. coli*		Absent	

Abbreviations: IR, infrared spectrophotometry; HPLC, high-performance liquid chromatography; Ph.Eur., European Pharmacopoeia; CRS, chemical reference substance; AV, acceptance value; CDCA, chenodeoxycholic acid; DCA, deoxycholic acid; MCA, methyl-cholic acid; FDA, U.S. Food and Drug Administration; TAMC, total aerobic microbial count; TYMC, total combined yeasts/molds count; *E. coli*, *Escherichia coli*; ^†^ Dose of the active substance is ≥25 mg and ≥25%. * Dose of the active substance is <25 mg or <25 per cent. ^±^ Time within the amount of dissolved cholic acid is ≥95% of the labelled content.

**Table 2 pharmaceutics-15-00773-t002:** Weight variation of 25 mg and 250 mg CA capsules (n = 60).

Capsule Strength (mg)	Minimum (mg)	Maximum (mg)	Mean (mg)	Standard Deviation
25 mg	196.3	220.2	209.9	0.0050
250 mg	235.4	263.6	251.0	0.0068

**Table 3 pharmaceutics-15-00773-t003:** Results of the product validation of 25 mg and 250 mg CA capsules.

	Batch Number					
	1	2	3	4	5	6
Strength	25	25	25	250	250	250
Batch size	1300	1300	1300	1200	1200	1200
Appearance	Complies	Complies	Complies	Complies	Complies	Complies
Identification	Complies	Complies	Complies	Complies	Complies	Complies
Content (% of labelled content)	94.7 ^¥^	90.6 ^¥^	95.4 ^¥^	102.6	102.2	100.7
Content uniformity						
Mass variation ^†^ (%)	−2.1–3.6	−4.1–2.8	−3.2–2.4	−3.3–4.3	−6.2–5.07	−5.5–3.8
Content uniformity * (AV)	7.3	8.2	14.7	5.8	10.8	6.4
Loss on drying (%)	0.70	0.80	0.80	0.10	0.20	0.20
Related substances (%)						
CDCA	0.00	0.00	0.00	0.00	0.00	0.00
DCA	0.00	0.00	0.00	0.00	0.00	0.00
MCA	0.09	0.10	0.10	0.14	0.13	0.10
Any unidentified substance	0.00	0.00	0.00	0.00	0.00	0.00
Total unidentified substances	0.00	0.00	0.00	0.00	0.00	0.00
Total Related Substances (%)	0.09	0.10	0.10	0.14	0.13	0.10
Dissolution (minutes) ^±^	5	5	5	10	10	10
Disintegration time (minutes)	5	6	6	9	8	8
Microbiology						
TAMC (CFU/g)	<5	<5	10	<5	<5	<5
TYMC (CFU/g)	<5	<5	<5	<5	<5	<5
*E. coli*	Absent	Absent	Absent	Absent	Absent	Absent

Abbreviations: AV, acceptance value; CDCA, chenodeoxycholic acid; DCA, deoxycholic acid; MCA, methyl-cholic acid; TAMC, total aerobic microbial count; TYMC, total combined yeasts/molds count; *E. coli*, *Escherichia coli*; Ph.Eur., European Pharmacopoeia; API, active pharmaceutical ingredient. ^¥^ Results were declared invalid due to low recovery. ^†^ Dose of the active substance is ≥25 mg and ≥25%. * Dose of the active substance is <25 mg or <25 per cent. ^±^ Time within the amount of dissolved cholic acid is ≥95% of the labelled content.

**Table 4 pharmaceutics-15-00773-t004:** CA and MCA content (% of labeled content) over time, stored under long-term conditions (25 °C ± 2 °C/60% ± 5% RH) and accelerated conditions (40 °C ± 2 °C/75% ± 5% RH).

Strength	Time Stored (Months)	Storage Condition	CA Content	MCA Content		
			**Mean (%) ± SD (n = 3)**	**Minimum (%)**	**Maximum (%)**	**Mean (%) ± SD** **(n = 3)**
25 mg	0	Long-term	NA ^¥^	0.09	0.10	0.01 ± 0.01
	3	Long-term	NA ^¥^	0.13	0.14	0.14 ± 0.01
	6	Long-term	100.03 ± 4.71	0.15	0.18	0.16 ± 0.02
	9	Long-term	93.40 ± 1.65	0.15	0.17	0.16 ± 0.01
	12	Long-term	94.43 ± 1.14	0.13	0.14	0.14 ± 0.01
	3	Accelerated	NA ^¥^	0.13	0.15	0.14 ± 0.01
	6	Accelerated	101.13 ± 0.83	0.16	0.17	0.17 ± 0.01
250 mg	0	Long-term	101.83 ± 1.00	0.10	0.14	0.12 ± 0.02
	3	Long-term	99.63 ± 0.64	0.21 *	0.27 *	0.23 ± 0.03
	6	Long-term	100.67 ± 0.64	0.17	0.19	0.18 ± 0.01
	9	Long-term	97.13 ± 1.04	0.13	0.16	0.15 ± 0.02
	12	Long-term	96.90 ± 2.71	0.13	0.14	0.14 ± 0.01
	3	Accelerated	99.60 ± 0.69	0.21 *	0.23 *	0.22 ± 0.01
	6	Accelerated	99.87 ± 1.35	0.11	0.14	0.13 ± 0.02

Abbreviations: RH, relative humidity; CA, cholic acid; MCA, methyl-cholic acid; SD, standard deviation; NA, data not applicable. Th measured content was below the specification limit (90.0–110.0% of labelled content). ^¥^ Results were declared invalid due to low recovery. * Above the specification limit (≤0.20%).

## Data Availability

Not applicable.

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
