# Peer review of "Product Validation and Stability Testing of Pharmacy Compounded Cholic Acid Capsules for Dutch Patients with Rare Bile Acid Synthesis Defects"

_pharmaceutics, 2023, doi:10.3390/pharmaceutics15030773_

Round 1
Reviewer 1 Report
Manuscript ID pharmaceutics-2163972 authored by Dr. Yasmin Polak et al is an interesting research aimed to describe pharmacy compounding of cholic acid containing capsules as an alternative to orphan drug. There are some issues that should be addressed before publishing consideration:
1. The documented description of superiority of cholic acid on BASDs treatment versus other molecules is missing
2. Particle size of API is very important in order to achieve an reproducible biopharmaceutical profile.
3. In previously published paper (doi: 10.3389/fphar.2021.758210) you developed CA containing capsules ( 25 mg and 250 mg potencies). Those formulas contained magnesium stearate. You should explain why this time you chose not to use the lubricant.
4. Manual filling of capsules leads to inter-batch/sub-batch variations and the quality of the finished product is not reproducible
5. Manual mixing of powders is not reproducible and can affect powder flow and in the end biopharmaceutical profile of the finished product
6. Validation of in-house developed methods is not shown. For example HPLC-RID methods used for quantification of CA during dissolution and related substances determination is missing. Also it is not clear which is the composition of the mobile phase (for both methods)
7. It is not clear how the CA capsules were packed in order to perform stability studies
8. Stability studies should result in an shelf life. The shelf life is not presented
9. In rows 229-230 you mention an optimized analysis method. This method is not presented
Reviewer 2 Report
1. In the abstract, mention stability testing conditions (temperature and humidity). The authors can include stability testing conditions in the text, i.e., For the stability study the capsules were analyzed at 0, 3, 6, 9 and 12 months.
2. Line 90, replace ‘ICH Q2 guidelines’ with ‘ICH Q2(R1) guideline’. Add a reference for this.
3. Line 95, replace ‘ICH Q1A guidelines’ with ‘ICH Q1(R2) guideline’.
4. In lines 235-236, the Authors state that ‘MCA is the methyl ester of CA that is normally produced by the liver’. It is not clear why have authors quoted this information. Add a reference to support this information.
5. Did authors test CA capsules of both strengths in open packs at long-term and/or accelerated stability conditions to determine the ‘in-use stability’ of the product?
6. Authors should provide overlaid chromatograms for related substance tests of CA capsule of both strengths (In supplementary section).
7. Line 82, correct the sentence, i.e., Column temperature was kept at 30˚C ± 2˚C.
8. Why have authors used two different column lengths (250 mm and 150 mm) for ‘assay & dissolution’ and ‘related substance’ tests. I am curious to know, if any specific reason.
Reviewer 3 Report
This article under review is well planned and written. Within the scope of the presented research, it contains all the necessary information. The performed validation procedures are well documented and confirm the Authors' conclusions. The development of a formulation for patients with rare diseases is undoubtedly a very important therapeutic problem. From this point of view, the publication of this article may be important to draw the attention of the medical community to these types of problems.
My doubts about publication in Pharmaceutics concern the aspects of scientific novelty in the presented article. It describes the standard procedures required for the quality control of a medicinal product.
Some interesting aspects from the scientific point of view appear in the article, such as an increase of the related substance MCA during 3 months of storage, during long term as well as accelerated conditions, which does not occur after 6 or 12 months. However, this aspect is not developed by the Authors experimentally. Bearing in mind the conclusions from EMA report they suggest to broad the specification limit for MCA.
Minor:
The work also does not specify where the standards for the determinations of the related substances, i.e. DCA and MCA, were obtained.
Round 2
Reviewer 1 Report
Dear authors,
You responded in a clear manner to all my major concerns. In my opinion the manuscript became fair for publication.